# Magnetization dependent tunneling conductance of ferromagnetic barriers

Zhe Wang [1,2✉], Ignacio Gutiérrez-Lezama[2,3], Dumitru Dumcenco [2], Nicolas Ubrig [2,3], Takashi Taniguchi [4], Kenji Watanabe [5], Enrico Giannini [2], Marco Gibertini [6,7] & Alberto F. Morpurgo [2,3✉]

Recent experiments on van der Waals antiferromagnets have shown that measuring the temperature ($T$) and magnetic field ($H$) dependence of the conductance allows their magnetic phase diagram to be mapped. Similarly, experiments on ferromagnetic $CrBr_3$ barriers enabled the Curie temperature to be determined at $H = 0$, but a precise interpretation of the magnetoconductance data at $H \neq 0$ is conceptually more complex, because at finite $H$ there is no well-defined phase boundary. Here we perform systematic transport measurements on $CrBr_3$ barriers and show that the tunneling magnetoconductance depends on $H$ and $T$ exclusively through the magnetization $M(H, T)$ over the entire temperature range investigated. The phenomenon is reproduced by the spin-dependent Fowler–Nordheim model for tunneling, and is a direct manifestation of the spin splitting of the $CrBr_3$ conduction band. Our analysis unveils a new approach to probe quantitatively different properties of atomically thin ferromagnetic insulators related to their magnetization by performing simple conductance measurements.

[1] MOE Key Laboratory for Nonequilibrium Synthesis and Modulation of Condensed Matter, Shaanxi Province Key Laboratory of Advanced Materials and Mesoscopic Physics, School of Physics, Xi'an Jiaotong University, Xi'an 710049, China. [2] Department of Quantum Matter Physics, University of Geneva, 24 Quai Ernest Ansermet, CH-1211 Geneva, Switzerland. [3] Group of Applied Physics, University of Geneva, 24 Quai Ernest Ansermet, CH-1211 Geneva, Switzerland. [4] International Center for Materials Nanoarchitectonics, National Institute for Materials Science, 1-1 Namiki, Tsukuba 305-0044, Japan. [5] Research Center for Functional Materials, National Institute for Materials Science, 1-1 Namiki, Tsukuba 305-0044, Japan. [6] Dipartimento di Scienze Fisiche, Informatiche e Matematiche, University of Modena and Reggio Emilia, IT-41125 Modena, Italy. [7] Centro S3, CNR-Istituto Nanoscienze, IT-41125 Modena, Italy. ✉email: zhe.wang@xjtu.edu.cn; alberto.morpurgo@unige.ch

Probing magnetism in atomically thin van der Waals crystals is challenging because most experimental methods commonly employed to study bulk compounds are not sufficiently sensitive to detect any magnetic signal from such a small amount of material[1–6]. Recently, it has been shown that magnetic phase boundaries (and even the complete magnetic phase diagram) of insulating atomically thin magnets can be detected by using them as tunnel barriers, and measuring their temperature-dependent magnetoconductance[7–16]. The sensitivity of the tunneling magnetoconductance to magnetism originates from the dependence of the tunneling probability on the magnetic state[17]. As $H$ and $T$ are varied across a magnetic transition, the alignment of the spins in the barrier changes sharply, and so does the tunneling probability of electrons with different spin orientations. The net result is an equally sharp change in the measured conductance that can be traced to identify the phase boundary.

These conclusions have been drawn from experiments on different antiferromagnetic insulators ($CrI_3$[7–10], $CrCl_3$[11–14] and $MnPS_3$[16]) that exhibit well-defined phase boundaries in the $H - T$ plane (i.e., in their phase diagram) associated to either spin-flip or spin-flop transitions, across which the symmetry of the magnetic state changes. The case of ferromagnetic insulators such as $CrBr_3$ (which has also been investigated by tunneling magnetotransport[14,15,18]) is conceptually subtler, because the underlying physics at zero or finite applied magnetic field is different. At $H = 0$, the critical temperature $T_C$ separates the paramagnetic state with a high rotational symmetry and vanishing magnetization ($M = 0$) from the ferromagnetic state with broken rotational symmetry and finite magnetization $M$. However, in the presence of any finite applied magnetic field $H$ the magnetization is finite ($M \neq 0$) at all temperatures (i.e., above or below $T_C$), and spin-rotational symmetry is always broken. As a result, there is no true continuous phase boundary separating the paramagnetic and ferromagnetic states in the $H - T$ plane (since the two states cannot be differentiated in terms of their symmetry)[19], the Curie temperature $T_C$ is an isolated point, and what precise information can be extracted from magnetotransport measurements is less obvious.

Earlier investigations of tunneling magnetotransport across $CrBr_3$ barriers[14,15,18] started to address these issues. It was shown that a peak in the resistance measured at zero applied magnetic field $H = 0$ allows the critical temperature $T_c$ to be determined. The evolution of the position of this peak upon increasing the applied magnetic field was employed to define heuristically a boundary between ferromagnetic and paramagnetic states, as done also in other systems. These experiments further showed that as the temperature is lowered well below $T_C$, the influence of the applied magnetic field on transport becomes increasingly weaker, and at $T = 4$ K only a featureless few percent change in resistance is observed as $H$ is increased up to a few Tesla.

Here we go beyond these results, and show directly from the experimental data that the temperature and magnetic field dependence of the tunneling conductance of $CrBr_3$ barriers is fully determined, at a quantitative level, by the magnetization of the material $M(H, T)$, throughout the entire temperature range investigated (spanning from well above to well below the critical temperature). The phenomenon is a direct consequence of the spin splitting of the conduction band in the material, which is governed by the magnetization, irrespective of whether magnetization occurs spontaneously due to ferromagnetism, or whether it is induced by the application of an external magnetic field (e.g., when the material in the paramagnetic state). We discuss how the dependence of the tunneling magnetoconductance on magnetization can be used to investigate different physical phenomena, such as critical fluctuations in the neighborhood of the Curie temperature or different microscopic models of magnetism in the material, which determine how the magnetization $M$ depends on applied field and temperature.

## Results

$CrBr_3$ is a van der Waals layered material that–irrespective of thickness (i.e., from bulk down to monolayer)– exhibits a transition to a ferromagnetic state with an easy axis perpendicular to the layers[20–28] (see Fig. 1a and b, Supplementary Note 1 and Supplementary Fig. 1,2). Bulk magnetization measurements in Fig. 1b show that the Curie temperature of our crystals is $T_C \simeq 32$ K, with a saturation magnetization corresponding to a magnetic moment of $3\mu_B$ per chromium atom, in line with previous reports[20–23]. Single crystals are exfoliated into thin layers and used to nano-fabricate hBN-encapsulated graphene/$CrBr_3$/graphene tunnel junctions inside a glove box (see inset of Fig. 1c for a schematic, Supplementary Fig. 3 for an optical image of the device, and Methods for detailed information about device assembly). Fig. 1c presents the current–voltage ($I$–$V$) characteristics for two representative devices with different thickness $d$ (corresponding to $N = 7$ and 8 layers), showing typical tunneling transport at low temperatures and the scaling behavior predicted by the Fowler–Nordheim (FN) tunneling formula (i.e., $\log(I/V^2) \propto 1/V$)[29,30]. As shown in Fig. 1d, the application of an external magnetic field (perpendicular to the $CrBr_3$ $ab$-plane) up to 3 T at $T = 2$ K causes only minor (<6%) and featureless variations in the conductance $G = I/V$. This is consistent with previous reports[14,15] and expected for a ferromagnetic semiconductor, in which the spins are spontaneously fully polarized already in the absence of an applied magnetic field at low temperature.

Despite the negligible low-temperature magnetoconductance, Fig. 1e shows that the conductance measured at zero applied magnetic field $G(H = 0, T)$ increases by a factor of three as $T$ is lowered from the Curie temperature $T_C = 32$ K down to 2 K (also in line with previous reports[14,15]). As the conductance is virtually temperature independent above $T_C$ (for $32 < T < 50$ K), we infer directly from the experimental data that the conductance increase is due to $CrBr_3$ entering the ferromagnetic state. This observation implies that magnetism does influence the electrical conductance of the tunnel barriers and that the effect is sizable: a threefold increase in conductance is comparable to the magnetoresistance of most common magnetic tunnel junctions (i.e., tunneling spin valve devices[31]) and of $CrCl_3$ antiferromagnetic tunnel barriers[11–15].

This observation motivates us to look more in detail at the temperature-dependent magnetoconductance of $CrBr_3$ barriers, $\delta G(H, T) \equiv [G(H, T) - G(0, T)]/G(0, T)$. The full dependence of $\delta G(H, T)$ on $T$ and $H$ is shown in Fig. 2a for both the 7- and 8-layer $CrBr_3$ devices, with the two of them exhibiting identical behavior. In both devices, $\delta G(H, T)$ is positive and peaks at $T = T_C$. The positive magnetoconductance can be understood, since at high $T$ the application of a magnetic field does lead to a better alignment of the spins in $CrBr_3$, causing the conductance to increase. We note that when $T$ approaches $T_C$ from above (i.w., coming from the paramagnetic state of $CrBr_3$) the magnetic field required to increase the conductance systematically decreases, indicating that the spin susceptibility $\chi$ is enhanced. This trend is reminiscent of the behavior expected in the critical regime in the neighborhood of the ferromagnetic transition[19].

The idea that the magnetoconductance for $T > T_C$ probes the critical regime of the paramagnetic state can be tested quantitatively if we recall that in a mean-field description, the linear spin polarizability $\chi \propto 1/(T - T_C)$ as $T$ approaches $T_C$[19]. We can then check whether the conductance depends on the magnetic field induced spin polarization, or equivalently on the magnetization (whose mean-field expression is given by the Curie–Weiss law,

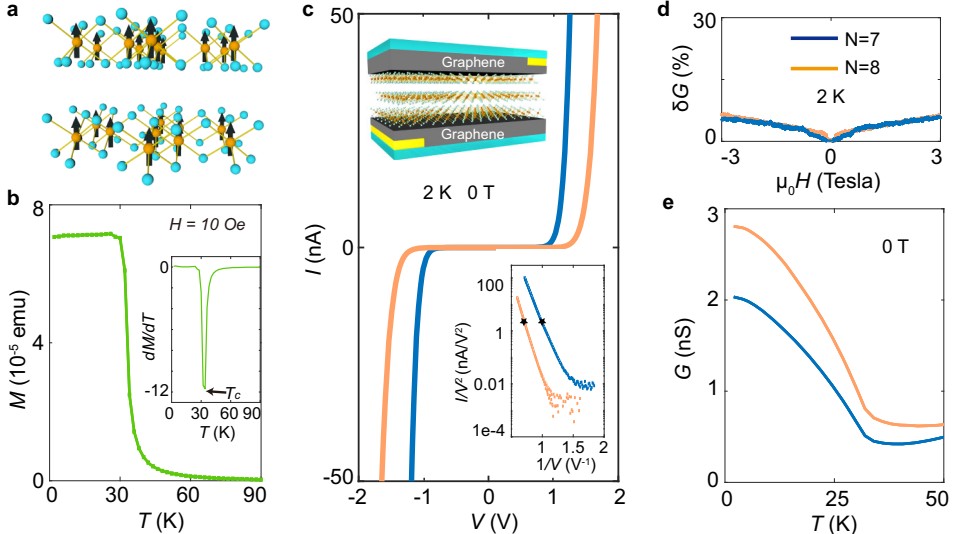

**Fig. 1 Tunneling conductance of CrBr₃ multilayers. a** Schematics of the crystal structure of CrBr₃: the orange balls represent the Cr atoms with the associated spins pointing perpendicularly to the layers; the light blue balls represent the Br atoms. **b** Temperature dependence of the magnetization measured on bulk CrBr₃ crystals with a magnetic field of 1 mT applied perpendicular to the layers. The inset shows the plot of $dM/dT$, with a sharp minimum close to 32 K, corresponding to the Curie temperature of CrBr₃. **c** Tunneling current as a function of applied voltage measured on a 7 (blue curve) and 8 (orange curve) layer CrBr₃ device at $T = 2$ K (curves of the same color in panels **d** and **e** represent data measured on the same devices). The up left inset is a cartoon representation of our hBN-encapsulated graphene/CrBr₃/graphene tunnel junction devices. The down right inset shows that -for sufficiently large applied bias $V$- $\log(I/V^2)$ is linearly proportional to $1/V$, as expected in the Fowler–Nordheim tunneling regime. To avoid Joule heating problems, we used relatively small bias voltage (as discussed in ref. [15]; $V = 1$ V and $V = 1.4$ V for the 7 and 8 layer devices, respectively) to measure the temperature and magnetic field dependence of the conductance shown in this and later figures, represented by the stars in the inset. **d** Magnetoconductance $\delta G(H, T) \equiv [G(H, T) - G(0, T)]/G(0, T)$ measured at $T = 2$ K on the 7 and 8 layer devices. The magnetic field is applied perpendicular to the CrBr₃ ab-plane. **e** Temperature dependence of the zero-field tunneling conductance of the 7 and 8 layer devices, exhibiting an increase of approximately 300%, as $T$ is lowered from $T_c$ to 2 K.

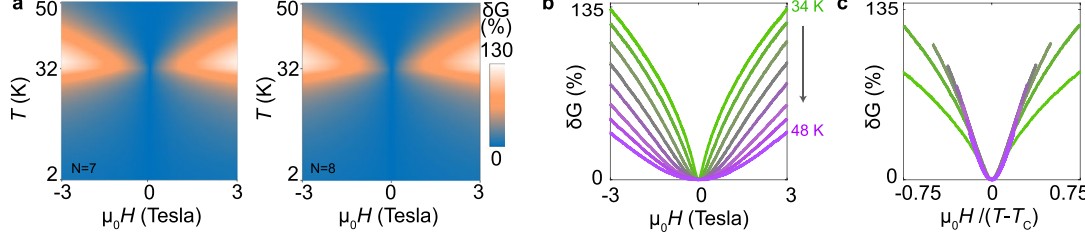

**Fig. 2 Temperature evolution of the tunneling magnetoconductance. a** Color plot of tunneling magnetoconductance for the 7 (left) and 8 layer (right) device as a function of applied magnetic field $\mu_0 H$ (with interval of 0.005 T) and temperature $T$ (with interval of 2 K). The bias voltage is 1 V and 1.4 V for 7 and 8 layer device, respectively. At fixed applied field, the magnetoconductance is maximum close to $T_c = 32$ K. **b** Magnetoconductance of seven layer device for $T > T_c$, as $T$ is varied from 34 to 48 K, in 2 K steps. **c**: when plotted as a function of $\mu_0 H/(T - T_C)$, the magnetoconductance curves shown in panel **b** collapse on top of each other at small field, indicating that in the linear regime the magnetocoductance $\delta G(H, T)$ depends on $H$ and $T$ only through the magnetization $M(H, T)$. In panels **b** and **c**, curves of the same color correspond to measurements done at the same temperature.

$M \propto \mu_0 H/(T - T_C)$), by simply plotting $\delta G(H, T)$ as a function of $\mu_0 H/(T - T_C)$ for any $T > T_C$ (see Fig. 2b). For sufficiently small $\mu_0 H/(T - T_C)$ all curves indeed collapse on top of each other (Fig. 2c), irrespective of the temperature at which they are measured, confirming that in the linear regime the field-induced increase of the conductance is determined by the net spin polarization (i.e., by the field-induced magnetization).

The relation between magnetoconductance and magnetization can be tested beyond the linear regime, by using the magnetization $M(H, T)$ measured on bulk crystals to re-plot the magnetoconductance of our tunnel barriers $\delta G(H, T)$ (Fig. 3a and b) as a function of $M$. The result is shown in Fig. 3c, with curves of different colors representing magnetoconductance measurements done at different temperatures. When plotted as a function of $M$

all curves collapse on top of each other throughout the entire range of $H$ and $T > T_C$ investigated. We can therefore conclude directly from the data that the magnetoconductance $\delta G$ depends on $H$ and $T$ only through the magnetization $M(H, T)$ even well outside the linear regime. That is: for $T > T_C$, $\delta G(H, T) = \delta G(M(H, T))$.

To extend our analysis from the paramagnetic state to $T < T_C$, when the CrBr₃ barriers are ferromagnetic, we look at the temperature dependence of the conductance measured at zero applied field. To this end, we consider the quantity $\Delta G(H, T) \equiv [G(0, T) - G(0, T_C)]/G(0, T_C)$, i.e., the relative increase in conductance observed as $T$ is lowered below the Curie temperature. If the conductance is a function of magnetization, this function should be the same underlying the behavior of $\delta G(H, T)$ for

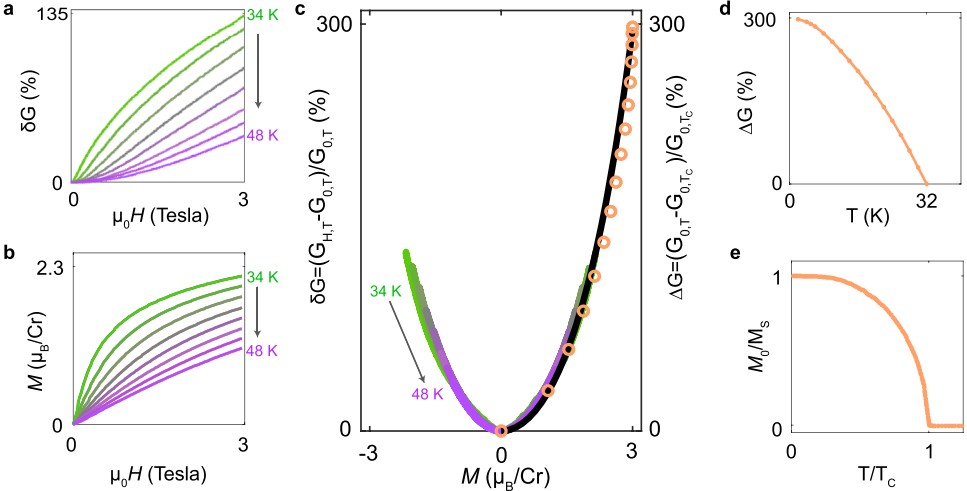

**Fig. 3 Magnetization dependence of the tunneling magnetoconductance. a** Tunneling magnetoconductance of 7 layer device measured for $T > T_c$, as $T$ is varied from 34 K to 48 K in 2 K steps (curves of the same color in panels **b** and **c** represent measurements taken at the same temperature). **b** Magnetic field dependence of the magnetization measured on bulk $CrBr_3$ crystals. **c**: Magnetization dependence of tunneling magneto-conductance. Colored lines represent the magnetoconductance measured at different $T > T_c$, plotted as a function of the bulk magnetization measured at the same temperature. The orange open circles represent the relative change in conductance due to the increase in the spontaneous magnetization of $CrBr_3$, measured for different $T < T_c$, obtained from the data shown in panels **d** and **e**. All data collapse on top of each other, indicating that the conductance is a function of the magnetization, i.e., it depends on $H$ and $T$ exclusively through $M(H, T)$, throughout the entire $T$ range investigated (i.e., from well below to well above $T_C$). The black line is a fit based on the expression obtained from the spin-dependent Fowler–Nordheim tunneling model, under the assumption that the splitting between the spin up and down bands is proportional to the magnetization (see main text for details). **d** Temperature dependence of the relative conductance increase as $T$ is lowered below $T_C$, in the ferromagnetic state of $CrBr_3$. **e** Spontaneous magnetization of $CrBr_3$ calculated by XXZ model with anisotropic exchange interactions that -as shown in ref. [24]- accurately reproduces the measured magnetization of atomically thin $CrBr_3$ crystals.

$T > T_C$, because the temperature dependence of $G(0, T)$ originates exclusively from the temperature dependence of the spontaneous magnetization $M(H = 0, T)$, which in the ferromagnetic state increases from zero at $T = T_C$, to its saturation value for $T \ll T_C$. Consistently with this idea, the data in Fig. 1e shows an increase in conductance upon lowering $T$. However, to establish whether the functional dependence of the magnetoconductance on magnetization below $T_C$ is the same as the one found for $T > T_C$ a more quantitative analysis is needed.

For such an analysis we cannot rely on the magnetization measured on bulk crystals. Indeed, in the ferromagnetic state for $T < T_c$ the magnetization of bulk $CrBr_3$ samples at $H = 0$ exhibit virtually no hysteresis and remnant magnetization (see Supplementary Fig. 2)[19,32], whereas in exfoliated atomically thin $CrBr_3$ crystals of the size used in our devices a clear hysteresis loop is observed, with finite remnant magnetization[24,27,28]. That is why in what follows we use the temperature-dependent spontaneous magnetization obtained in Hall magnetometry experiments that–as discussed in ref. [24]–is very well reproduced by the temperature dependence calculated in a mean-field treatment of the XXZ model with anisotropic exchange interaction. Specifically, we re-plot the quantity $\Delta G(H, T)$ shown in Fig. 3d, as a function of $M$ using the spontaneous magnetization curve shown in Fig. 3e (which we have reproduced from ref. [24]). The result is represented by the open circles in Fig. 3c. The data fall on top of the $\delta G(M)$ curve found in our analysis of transport in the paramagnetic state of $CrBr_3$, for $T > T_C$.

The excellent agreement demonstrates that the conductance of $CrBr_3$ tunnel barriers depends on temperature and magnetic field only through its magnetization over the full experimental range investigated, and that the dependence is described by the same function from well above $T_C$ to the lowest temperature reached in our measurements (2 K). The conclusion is extremely robust, because it is drawn directly from the analysis of the experimental data, without any theoretical assumption (the 8-layer device

exhibits an identical behavior, as discussed in Supplementary Note 2 and shown in Supplementary Fig. 4).

The observed dependence of the magnetoconductance on the magnetization is different from that reported in tunneling experiments on antiferromagnetic tunnel barriers. Indeed, the conductance of $CrI_3$ multilayer barriers depends on the precise magnetic state of the multilayer, i.e., on the precise magnetization configuration in the different layers, and not only on the total magnetization[7]. Flipping the magnetization of a layer deep inside the tunnel barrier, for instance, change the conductance much more than flipping the magnetization of outer layer next to the injecting contact. In $MnPS_3$ multilayer barriers, the conductance actually decreases when the magnetization starts increasing[16] (past the spin-flop transitions that occurs in this material), a behavior opposite to that seen in tunneling magnetotransport through $CrBr_3$. These considerations, together with our discussion below, indicate that a positive tunneling magnetoconductance depending exclusively on magnetization $M(H, T)$ is a distinctive properties of ferromagnetic barriers. This is a non-trivial conclusion because it is not obvious a priori that–even for ferromagnets–the tunneling magnetoconductance should depend only on $M$, which describes the average degree of alignment of the spins. If spin fluctuations were important for tunneling transport, for instance, the conductance could be expected to depend not only on the magnetization, but also on the magnetic susceptibility. Our experiments show that this is not the case.

## Discussion
The experimental results reported above have a straightforward interpretation within the context of FN tunneling transport commonly used to interpret the conductance of van der Waals magnetic barriers. In FN tunneling, the applied bias tilts the conduction band across the $CrBr_3$ layer, effectively reducing the thickness of the tunnel barrier, so that eventually the tunneling

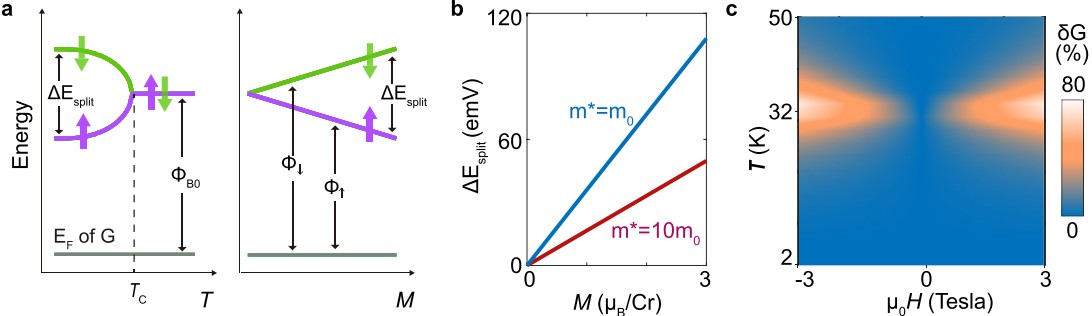

**Fig. 4 Relation between spin splitting energy and magnetization. a** Schematic diagram of the relevant energies involved in the electron tunneling process. The spin-up and spin-down CrBr₃ conduction bands become nondegenerate when the magnetization $M$ is finite, causing a different height of the tunneling barrier for electrons whose spin points in opposite directions (the barrier height is given by the distance between the corresponding band edge and the Fermi level $E_F$ in the graphene electrodes, indicated by the bottom horizontal line). A finite $M$ can be induced by the appearance of the spontaneous magnetization as $T$ is lowered below $T_C$, or by the application of an external magnetic field for $T > T_C$. **b** Magnetization dependence of the splitting energy, as determined from fitting of the data in Fig. 3c using the spin-dependent Fowler–Nordheim tunneling model to model the conductance. The blue and purple lines correspond to the results obtained assuming the effective mass in the direction perpendicular to the layers of CrBr₃ to be $m_0$ or $10m_0$, respectively. **c** Color map of magnetoconductance calculated using Eq. (2) in the main text, using the Weiss model to determine the dependence $M(H, T)$ of the magnetization on magnetic field and temperature.

probability for electrons becomes sizable and a finite current is observed[29,30]. The $I − V$ characteristics in the FN tunneling regime satisfy the relation:

$$I \propto \frac{V^2}{\phi_B} e^{\frac{-8\pi d\sqrt{2m^*}\phi_B^{3/2}}{3heV}}, \quad (1)$$

where $m^*$ is the effective mass describing the motion of electrons in CrBr₃ in the direction perpendicular to the planes, $\phi_B$ is the barrier height determined by the distance between the Fermi level in the contacts and the conduction band edge in CrBr₃, $h$ is Planck's constant and $e$ the (modulus of the) electron charge. For a ferromagnet, an analogous relation is expected to hold separately for electrons with spin up and spin down, which experience different barrier heights $\phi_\uparrow$ and $\phi_\downarrow$, due to the spin-splitting of the conduction band present for $T < T_C$ (see Fig. 4a). The total conductance is then the sum of the contributions given by electrons with spin up and spin down:

$$G = G^\uparrow + G^\downarrow = A \frac{V}{\phi_\uparrow} e^{\frac{-8\pi d\sqrt{2m^*}\phi_\uparrow^{3/2}}{3heV}} + A \frac{V}{\phi_\downarrow} e^{\frac{-8\pi d\sqrt{2m^*}\phi_\downarrow^{3/2}}{3heV}}, \quad (2)$$

where A is a constant determined by the barrier dimensions.

We use this expression to analyze the experimental data by assuming that the spin splitting of the conduction band in the ferromagnetic state is linearly proportional to the magnetization(see Fig. 4b), resulting in barrier heights for spin up and down given by $\phi_{\uparrow,\downarrow} = \phi_{B0} \pm \gamma M$. We calculate the magnetoconductance $[G(M) − G(M=0)]/G(M=0)$ using the value of $\sqrt{2m^*}\phi_{B0}$ extracted from the measured $I − V$ curves, treating $\gamma$ as the sole fitting parameter. The result of this procedure, represented by the black curve in Fig. 3c, reproduces the experimental data perfectly. Interestingly, a conceptually similar approach has been followed in earlier beautiful work on the tunneling conductance of EuO barriers in the ferromagnetic state[33]. That work, however, focused exclusively on the case of $T < T_C$ and zero applied magnetic field (by looking at the temperature dependence of the conductance in terms of the measured temperature dependence of the magnetization),and did not analyze the behavior for $T > T_C$. Our results show that the approach has a much broader validity: it can be applied both below and above $T_C$, it remains valid in the presence of a magnetic field, and since CrBr₃ and EuO are very different materials, it applies to very different classes of ferromagnetic insulators. Indeed, Eu is a rare earth element with $4f$ orbitals filled

by electrons and EuO is a covalently bonded material, whereas Cr is a transition metal in which magnetism originates from electrons filling $3d$ orbitals and CrBr₃ is a van der Waals bonded compound. Finding that two such different materials exhibit an analogous dependence of the tunneling magnetoconductance on the magnetization is worth noting.

If used in conjunction with a model predicting the magnetic field and temperature dependence of the magnetization, this approach allows the full magnetoconductance to be calculated. This is illustrated by the color plot in Fig. 4c that–despite having being obtained with the simplest possible Weiss model of Ising ferromagnetism–reproduces all the qualitative features observed in the experiments (compare Fig. 4c with Fig. 2a), and even exhibits a nearly quantitative agreement. Alternatively, it is also possible to extract the temperature and magnetic field dependence of the magnetization from the measured magnetoconductance, as discussed in Supplementary Note 2 and shown in Supplementary Fig. 4. The excellent agreement between the calculated and the measured magnetoconductance (see Fig. 3c) suggests the possibility to extract the spin splitting energy quantitatively, from the value of the fitting parameter $\gamma$. This is however not straightforward, because Eq. (2) depends on the product $\sqrt{m^*}(\phi_0)^{3/2}$, and the effective mass $m^*$ is not known. Fig. 4b shows the spin splitting energy as a function of magnetization obtained by taking the value of $\gamma$ used to fit the $\delta G = \delta G(M)$ curves in Fig. 3c, and assuming the effective mass $m^*$ to be either the free electron mass $m_0$ or 10 $m_0$, a very large value chosen to mimic the flatness of the CrBr₃ bands in the direction perpendicular to the layers[34,35]. We find that at saturation the energy splitting separating the spin-up and the spin-down bands is approximately 110 meV if $m^* = m_0$ and 50 meV if $m^* = 10m_0$, indicating that for realistic values of the effective mass the spin-splitting energy is between 50 and 110 meV. We emphasize, however, that care is certainly needed in interpreting the meaning of this quantity microscopically, because–as applied to CrBr₃ barriers–the FN model is a phenomenological approach that does not take into account the complexity of the microscopic electronic structure of the material. In particular, it does not take into account that the conduction band consists of two distinct, nearly degenerate electronic bands originating from the $e_{1g}$ and $t_{2g}$ orbitals of the Cr atoms.

The key result of our work is that the measured tunneling magnetoconductance of CrBr₃ is entirely determined by its magnetization, which is why magnetoconductance

measurements can be used to investigate the magnetic properties of the material. We envision that magnetoconductance measurements will allow detailed investigations of the critical behavior of the magnetic susceptibility in the paramagnetic state for $T$ very close to $T_C$ and provide a new, experimentally simple way to determine critical exponents. This is possible because the required data analysis only relies on the fact that at small $M$ the magnetoconductance is a quadratic function of the magnetization (see Supplementary Note 3 and Supplementary Fig. 5). Another interesting possibility is to analyze magnetoconductance measurements over a broad range of temperatures and magnetic fields, to discriminate between microscopic theoretical models that predict a different functional dependence for $M(H, T)$ (see Supplementary Note 3 and Supplementary Fig. 6). These are just two examples that illustrate the most important aspect of our results, namely that measurements of the tunneling conductance are not limited to the investigation of antiferromagnetic barriers, but can also provide detailed information about the magnetic properties of ferromagnetic insulators.

## Methods

**Bulk crystal growth and characterizations.** Crystals of $CrBr_3$ were grown by the Chemical Vapor Transport method as reported earlier[36]. Pure Chromium (99.95% CERAC) and $TeBr_4$ (99.9% Alfa Aesar) were mixed with a molar ratio 1:0.75 to a total mass of 0.5 g, and put in quartz tube with an internal diameter of 10 mm and a length of 13 cm. The preparation of the quartz reactor was done inside a glove box under pure Ar atmosphere. The tube was evacuated down to $10^{-4}$ mbar and sealed under vacuum, then put in a horizontal tubular furnace in a temperature gradient of about 10°C/cm, with the hot end at 700°C and the cold end at 580°C. After seven days at this temperature, the furnace was switched off, and the tube cooled to room temperature. $CrBr_3$ was found to crystallize at the cold end of the tube. Shiny, thin platelet-like, dark green-blackish single crystals of typical lateral size of 1–5 mm were extracted. Bulk crystals of 2.1 mg were used for the magnetic characterization in a MPMS3 SQUID magnetometer (Quantum Design). The magnetic moment of the crystals was measured with magnetic field parallel to the crystallographic c-axis.

**Tunneling junction fabrication and transport measurements.** $CrBr_3$ multilayers were mechanically exfoliated from the crystals discussed in the section of crystal growth. Tunnel junctions of multilayer graphene/$CrBr_3$/multilayer graphene were assembled using a pick-and-lift technique with stamps of PDMS/PC. To avoid degradation of thin $CrBr_3$ multilayers, the exfoliation of $CrBr_3$ and the heterostructure stacking process were done in a glove box filled with Nitrogen gas, and the whole tunneling junction was encapsulated with hBN before being taken out. Conventional electron beam lithography, reactive-ion etching, electron-beam evaporation (10 nm/50 nm Cr/Ar) and lift-off process were used to make edge contacts to the multilayer graphene. The thickness of the layers was determined by atomic force microscope measurements performed outside the glove box, on the encapsulated devices. Transport measurements were performed in a cryostat from Oxford Instruments, using home-made low-noise electronics.

## Data availability

All relevant data are available from the corresponding authors upon reasonable and well-motivated request.

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

## Acknowledgements

We sincerely acknowledge Alexandre Ferreira for technical support. Z.W. acknowledges the National Natural Science Foundation of China (Grants no. 11904276). A.F.M. gratefully acknowledges financial support from the Swiss National Science Foundation (Division II) and from the EU Graphene Flagship project. M.G. acknowledges support from the Italian Ministry for University and Research through the Levi-Montalcini program. K.W. and T.T. acknowledge support from the Elemental Strategy Initiative conducted by the MEXT, Japan, Grant Number JPMXP0112101001 and JSPS KAKENHI Grant Number JP20H00354.

## Author contributions

Z.W. and A.F.M. conceived the work. D.D. and E.G. grew $CrBr_3$ crystals and performed bulk characterization. T.T. and K.W. provided high-quality boron nitride crystals. Z.W. fabricated samples and performed transport measurements with help of I.G. and N.U.. Z.W., I.G., N.U., M.G., and A.F.M. analyzed and interpreted the magnetoconductance data. All authors contributed to writing the paper.

## Competing interests

The authors declare no competing interests.
