## [Peer Review File · Nature Communications]

Magnetization dependent tunneling conductance of ferromagnetic barriersREVIEWER COMMENTS

Reviewer #1 (Remarks to the Author):

The authors report tunnelling conductance measurements of atomically thin CrBr₃, and study the external magnetic field and temperature dependence. The authors find that the magnetoconductance is determined by the magnetization, which can be a new way to study magnetic properties. The results are well presented, the paper is well written, and the story is supported by good-quality data. However, concerns and questions need to be addressed.

1. Main concern. There have been several studies on CrBr₃ using different probes, as the authors cited, including tunnelling conductance, Hall and NV magnetometry, STM, photoluminescence, and MCD. So the magnetic properties of CrBr₃ have been well understood. In particular, References 13 and 14 have reported similar results about the magnetic field and temperature dependence of the tunnelling conductance, and also studied the M-H phase diagram. The authors may want to discuss more how this study compares to these previous works.
2. The study mainly reports seven- and eight-layer CrBr₃ samples. Is there a reason the authors only focus on this thickness? What will happen to thinner samples?
3. On page 3, the authors say that “the devices behave as single domains”. However, the references cited here actually reported that there can be magnetic domains in atomically thin CrBr₃. In addition, how can the authors tell if there are magnetic domains in ferromagnetic CrBr₃ from their tunnelling conductance measurements here?
4. On page 5, the analysis of the spin splitting follows closely to the previous study on EuO, and the authors state the new finding here is that the approach is also valid above T_c, and CrBr₃ and EuO are “very different” materials. The authors may want to discuss why it has a broader validity and why CrBr₃ and EuO are very different here considering they are both ferromagnetic.
5. Minor point. The authors mention Fig. 4d in the main text, I guess they mean Fig. 4.c.

Reviewer #2 (Remarks to the Author):

In this work, the authors introduce a new method of using the tunneling conductance of a 2D insulating ferromagnet, CrBr₃, to extract information about its magnetic order. They argue that measuring conductance as a function of B and T is directly related to the magnetization both above and below the Curie temperature. In addition to identifying the Curie temperature of an ultrathin 2D ferromagnet, this technique also enables extraction of critical exponents near the phase transition as well as a method to validate different magnetization models for the material.

The family of 2D magnets continues to expand, including the exploration of the broad class of transition metal halides, which cannot be probed by direct transport measurements due to their insulating nature. As a result, tunneling magnetoconductance is a useful tool of broad interest to the 2D materials community to study insulating 2D magnets in the few-layer limit, which often differ from their bulk properties. I therefore recommend the publication of this work in Nature Communications.

Reviewer #3 (Remarks to the Author):

The authors have performed temperature-dependent magnetotransport measurements in CrBr₃ tunnel barrier devices. They analyzed the data to conclude the tunneling conductance in graphene-CrBr₃-graphene sandwich devices is exclusively dependent on the magnetization of CrBr₃. The main contribution of the paper shows a direct dependence of tunneling magnetoconductance on the magnetization of ferromagnetic insulators. Whereas analysis of this data is new, the same device has been investigated before in detail with very similar techniques in the references quoted in this manuscript (PNAS 116, 11131 (2019), Nano Lett. 19, 5739 (2019)). Whereas I think the experiments are detailed and carefully performed, and the data analysis is also sound, I do not agree with some of the major conclusions the authors have drawn in this manuscript. My concerns regarding the manuscript are elaborated below.

1. Page 1 abstract: "These findings demonstrate that the investigation of magnetism by tunneling conductance measurements is not limited to antiferromagnets, but can also be applied to ferromagnetic materials."

Page 1- paragraph 3- "it is not at all clear that the technique can be equally effective to probe ferromagnetic insulators since in that case, no magnetic phase boundaries are present below the Curie temperature."

I do not agree with these statements. In a magnetic tunnel barrier, be it ferromagnetic or antiferromagnetic, the tunneling conductance is expected to be dependent on magnetization, it would be rather surprising if the tunneling conductance would not reflect the magnetization state of the ferromagnetic insulator.

2. The main conclusion of the paper, as highlighted by the authors on page 5, last paragraph, "magnetoconductance measurements can be used to investigate the magnetic properties of the material." and "magnetoconductance measurements will allow detailed investigations of the critical behavior of the magnetic susceptibility in the paramagnetic state for T very close to T_C and provide a new, experimentally simple way to determine critical exponents". These need to be highlighted more in the abstract of the paper.

3. Page 1- paragraph 3- "These conclusions have been drawn from experiments on different antiferromagnetic insulators (CrI₃ [7–9], CrCl₃ [10–13] and MnPS₃ [15]), and it is not at all clear

that the technique can be equally effective to probe ferromagnetic insulators since in that case, no magnetic phase boundaries are present below the Curie temperature. Indeed, tunneling conductance measurements on ferromagnetic CrBr₃ barriers have shown only an extremely small and featureless low-temperature tunneling magnetoconductance^{13,14,17}"

Whereas I agree, that in previous literature, tunneling conductance measurements on ferromagnetic CrBr₃ barriers have shown only an extremely small and featureless low-temperature tunneling magnetoconductance, I argue that the experimental data for CrBr₃ presented in (Nano Lett. 2019, 19, 5739, reference 14 in the manuscript) has similar behavior compared to the present manuscript, and show that tunneling measurements allow for an easy determination of the field–temperature phase diagram.

4. Fig. 1c bottom right inset: It is more appropriate to plot this data as a function of $1/V$, as d is just a constant for both the curves. Plotting as a function of $1/V$ also makes it clear above which voltage the tunnel current turns on.

5. Fig. 1c main panel: it will be good to highlight which point in the I/V curve was chosen to plot 1e. Naively, from eye-estimate 1 V and 1.4 V seems to lie just above the turning point. How does the $G-T$ curve change if a higher current value is chosen?

6. Which precautions were taken to avoid Joule heating while performing low-temperature measurements?

7. Fig. 1d): $\delta G-T$ shows a weak-localization behavior. Can this be attributed to the graphite leads?

8. Fig. 2 a and b): Which temperature and magnetic field intervals were used to obtain these plots?

9. Fig 2: At which values of voltage or current were this data obtained?

10. Page 3, paragraph 2: "We can therefore conclude directly from the data that the magnetoconductance δG depends on H and T only through the magnetization $M(H; T)$ even well outside the linear regime. That is: for $T > T_C$, $\delta G(H; T) = \delta G(M(H; T))$ ". Can the authors discuss which other factors can potentially influence magnetoconductance in these systems, that are eliminated by this data? For example, CrBr₃ is known to influence Hall measurements in graphene through proximity coupling (Nature Electronics volume 2, pages457–463 (2019)).

11. Page 3, paragraph 4: "exfoliated atomically thin crystals of the size used in our devices have been found to behave as single domains". The movement of domain walls was previously reported in monolayer CrBr₃ (Nature Electronics volume 2, pages457–463 (2019)).

12. Is the data shown in Fig. 3 obtained in the 7-layer device? Is Fig. 3a just a repetition of Fig. 2b?

13. Methods: "Crystals of CrBr₃ were grown by the Chemical Vapour Transport method as reported earlier." Reference is missing here.

14. It is not absolutely necessary, but it will be nice to see images of grown crystals in the supporting information.

15. According to the images shown in supplementary Fig. 2: The heterostructures have quite a few bubbles trapped. Can the authors show a closer image of the tunnel junction area, to demonstrate no bubbles are present in that area? Or, even if the bubbles are preset, can the authors comment on how the bubbles are going to affect the magnetoconductance behavior?

Along with the above I also have a few minor comments:

1. In formal scientific writing "em dash" is not used widely. I request the authors to minimize the use of em-dash. In addition, the authors are requested to note that, no space is required before or after emdash.
2. Page 1 paragraph 4 line 13: "scheme" will be "schematic"
3. Page 5 paragraph 1: "This is illustrated by the color plot in Fig. 4d" and "compare Fig. 4d with Fig. 2a"- Here "4d" will be "4c"
4. Fig. 1: "insert" will be "inset"

Reviewer #4 (Remarks to the Author):

The authors present experimental results on the tunneling magnetoconductance measurements across graphene/CrBr₃/graphene devices. They analyze the magnetoconductance using the conductance at zero magnetization state ($H=0$, $T>=T_c$) as references. They find that the behavior of the normalized magnetoconductance solely depends on the magnetization of CrBr₃. This behavior can be explained by the Fowler-Nordheim model by taking into account the magnetization dependent tunnel barrier heights for the two spin channels. These results are interesting and useful for the 2D magnet community. I would recommend publication in Nature Communications. Below I have a few minor comments.

1. When analyzing the data below T_c , the authors use the XXZ model to calculate the expected M values instead of using the experiment results due to the domain formation at small fields (1 mT). Would it be possible to use the measured M values at slightly higher field, large enough to polarize the domain but small enough to induce any appreciable change in M , for example, 0.1 T? Also, the authors should provide a short description of the model calculation.
2. Fig. 3c is calculated based on fitting parameters obtained from the experimental data. But the maximum ΔG is only 80% while the maximum in experiment is 130% (Fig. 2a). What causes such a big difference?
3. The direction of the magnetic field in the magnetoconductance measurements should be specified somewhere in the paper.

Reply to the Referees

Reply to Reviewer #1

Comment 1: *The authors report tunnelling conductance measurements of atomically thin CrBr₃, and study the external magnetic field and temperature dependence. The authors find that the magnetoconductance is determined by the magnetization, which can be a new way to study magnetic properties. The results are well presented, the paper is well written, and the story is supported by good-quality data. However, concerns and questions need to be addressed.*

Reply 1: We are pleased to read that the referee thinks that our results are well presented and that our conclusions are supported by good quality data. We thank the reviewer for correctly summarizing the main point of the paper, namely that the tunneling magnetoconductance is determined by the magnetization and that therefore tunneling magnetoconductance measurements offer a new way to study the magnetic properties of atomically thin ferromagnetic insulators. The reviewer also has some concerns and questions, which we address point-by-point here below.

Comment 2: *Main concern. There have been several studies on CrBr₃ using different probes, as the authors cited, including tunnelling conductance, Hall and NV magnetometry, STM, photoluminescence, and MCD. So the magnetic properties of CrBr₃ have been well understood. In particular, References 13 and 14 have reported similar results about the magnetic field and temperature dependence of the tunnelling conductance, and also studied the M-H phase diagram. The authors may want to discuss more how this study compares to these previous works.*

Reply 2: We are aware of earlier work on CrBr₃ and we have referred to the papers that are relevant for our study. The main result of our work is not the magnetoconductance measurements themselves, which were reported earlier by others. What is new is the systematic and quantitative interpretation of the measurements as a function of temperature and magnetic field, which shows directly from the data that the tunneling magnetoconductance is fully determined by the magnetization. This is important for several reasons, including the fact—fully appreciated by the referee (see Comment 1)—that our results enable the magnetization of atomically thin ferromagnetic insulators to be investigated by means of magnetotransport measurements.

In the resubmitted manuscript we have added a paragraph describing earlier work on tunneling magnetotransport, indicating that these early measurements did already establish a number of facts (for instance, that magnetotransport allow the Curie temperature to be determined and showed that the low-temperature magnetoresistance is small). We have also discussed more explicitly and extensively the physical reason which make the interpretation of transport measurements on a ferromagnetic insulator such as CrBr₃ subtler, as compared to the interpretation of the same type of measurements on antiferromagnetic insulators. Since this point is important, we discuss it here as well in detail.

Interpreting magnetotransport measurements on antiferromagnetic insulators is conceptually simpler than in ferromagnetic insulators, because in antiferromagnets there are well-defined continuous boundaries in the H - T diagram—typically occurring at spin-flip and spin-flop transitions—separating phases that have different symmetry. In a ferromagnet such as CrBr₃ this is not the case. A true phase transition leading to a change in symmetry of the ground state only takes place at zero applied field

($H=0$). Indeed, at $H=0$ the magnetization M vanishes for $T > T_c$ and is finite for $T < T_c$, so that at T_c the symmetry of the system changes. However, at any finite applied magnetic field, the magnetization M is finite at all temperatures: there is no temperature at which the symmetry of the system changes sharply and therefore there is no precisely defined phase boundary (i.e., at finite applied magnetic field, the paramagnetic state and the ferromagnetic state cannot be discriminated in terms of symmetry). Of course, one can adopt –as it is often done– a heuristic criterion to define a critical temperature also at finite magnetic field. However, a critical temperature defined in this way does not necessarily correspond to any sharp change in the system properties and therefore also not in the magnetoconductance. This is why finding an unambiguous way to interpret the tunneling magnetotransport data in a ferromagnet is difficult. This is also why our finding that magnetotransport provide well-defined information about a physically measurable and unambiguously defined quantity such as the magnetization is relevant.

We have rewritten the introduction to explain this important point much more explicitly than in the originally submitted version of the manuscript.

Comment 3: *The study mainly reports seven- and eight-layer CrBr₃ samples. Is there a reason the authors only focus on this thickness? What will happen to thinner samples?*

Reply 3: To establish our conclusions, it is essential to be able to plot the measured magnetoconductance as a function of magnetization. In the paramagnetic state, the only way for us to do that is to use magnetization data measured on bulk crystals. For the analysis it is then essential that the multilayers have the same T_c as the bulk. The multilayers we chose (7 or 8 layer thick) ensure that this is the case. Thinner multilayers would cause problems, because their T_c is slightly suppressed, as found in the paper on Hall bar magnetometry by the Manchester group (indeed, note that for the analysis of magnetotransport for $T < T_c$ we use the data from Manchester for $M(T, H=0)$ plotted as a function of T/T_c , and use “our” T_c for the data analysis).

In short, we expect that thinner samples will exhibit the same physical behavior but the different T_c as compared to bulk crystals would prevent us to perform the analysis that we discuss in our manuscript.

Comment 4: *On page 3, the authors say that “the devices behave as single domains”. However, the references cited here actually reported that there can be magnetic domains in atomically thin CrBr₃. In addition, how can the authors tell if there are magnetic domains in ferromagnetic CrBr₃ from their tunnelling conductance measurements here?*

Reply 4: We agree with the referee that our phrasing has been unprecise. What we really meant is that magnetization measurements on bulk crystals show no hysteresis and no remnant magnetization at zero applied field, whereas in atomically thin layers a clear hysteresis cycle is observed and the remnant magnetization is non-zero. The difference is due to magnetic domains, but we agree with the referee that it is not precise to write –as we had done in the originally submitted version– that the atomically thin crystal behave as single domains.

In the resubmitted version of our manuscript we have modified the phrasing, and we now refer to hysteresis and remnant magnetization (and not to single domain behavior) to describe the difference in behavior of the magnetization of bulk and few layer crystals.

Comment 5: *On page 5, the analysis of the spin splitting follows closely to the previous study on EuO, and the authors state the new finding here is that the approach is also valid above T_c , and CrBr₃ and EuO are “very different” materials. The authors may want to discuss why it has a broader validity and why CrBr₃ and EuO are very different here considering they are both ferromagnetic.*

Reply 5: The two materials are very different because Europium is a rare earth material with electrons occupying the 4f shell, and its atomic number is large (which implies a strong spin orbit interaction). Also, EuO is a covalently bonded compound. In contrast, Cr is a much lighter 3d transition metal element, and CrBr₃ is a van der Waals bonded compound. It is therefore noteworthy that –despite these differences– the conclusion that magnetotransport depends only on magnetization appears to be valid in both compounds.

As for the broader validity of our finding, the fact that the magnetoconductance is a function of magnetization had been shown for EuO for $T < T_c$ at $H=0$ only. The work on EuO did not explore whether the conclusion is also valid for $T > T_c$ or at finite field. In our manuscript, we show that the magnetoconductance is a function of magnetization a very different material (see discussion here above), for T above as well as below T_c , and both in the absence and in the presence of an applied magnetic field. That is what we mean when we state that our work demonstrates the much broader validity of the relation between magnetconductance and magnetization.

We have modified the text in the resubmitted version of the manuscript to mention explicitly in which sense EuO and CrBr₃ are different materials.

Comment 6: *Minor point. The authors mention Fig. 4d in the main text, I guess they mean Fig. 4.c.*

Reply 6: We thank the reviewer for pointing it out and we have gone through the manuscript and corrected all errors of this type that we found.

Reply to Reviewer #2:

Comment 1: *In this work, the authors introduce a new method of using the tunneling conductance of a 2D insulating ferromagnet, CrBr₃, to extract information about its magnetic order. They argue that measuring conductance as a function of B and T is directly related to the magnetization both above and below the Curie temperature. In addition to identifying the Curie temperature of an ultrathin 2D ferromagnet, this technique also enables extraction of critical exponents near the phase transition as well as a method to validate different magnetization models for the material.*

The family of 2D magnets continues to expand, including the exploration of the broad class of transition metal halides, which cannot be probed by direct transport measurements due to their insulating nature. As a result, tunneling magnetoconductance is a useful tool of broad interest to the 2D materials community to study insulating 2D magnets in the few-layer limit, which often differ from their bulk properties. I therefore recommend the publication of this work in Nature Communications.

Reply 1: We are very pleased to read the positive comment of the referee and thank her/him for the recommendation of direct publication.

Reply to Reviewer #3:

Comment 1: *The authors have performed temperature-dependent magnetotransport measurements in CrBr₃ tunnel barrier devices. They analyzed the data to conclude the tunneling conductance in graphene-CrBr₃-graphene sandwich devices is exclusively dependent on the magnetization of CrBr₃. The main contribution of the paper shows a direct dependence of tunneling magnetoconductance on the magnetization of ferromagnetic insulators. Whereas analysis of this data is new, the same device has been investigated before in detail with very similar techniques in the references quoted in this manuscript (PNAS 116, 11131 (2019), Nano Lett. 19, 5739 (2019)). Whereas I think the experiments are detailed and carefully performed, and the data analysis is also sound, I do not agree with some of the major conclusions the authors have drawn in this manuscript. My concerns regarding the manuscript are elaborated below.*

Comment 1: We thank the reviewer for acknowledging that the analysis of the data in our manuscript (which is the core part of our results) is new and that both the analysis and the measurement are sound. The referee also has some concerns that regard a number of specific statements that we made in our manuscript. We address these concerns in the point-by-point reply below.

Comment 2: *Page 1 abstract: "These findings demonstrate that the investigation of magnetism by tunneling conductance measurements is not limited to antiferromagnets, but can also be applied to ferromagnetic materials."*

Page 1- paragraph 3- "it is not at all clear that the technique can be equally effective to probe ferromagnetic insulators since in that case, no magnetic phase boundaries are present below the Curie temperature."

I do not agree with these statements. In a magnetic tunnel barrier, be it ferromagnetic or antiferromagnetic, the tunneling conductance is expected to be dependent on magnetization, it would be rather surprising if the tunneling conductance would not reflect the magnetization state of the ferromagnetic insulator.

Reply 2: After reading the comments of the referee, we see that our statements may have been ambiguous and may have led themselves to misunderstandings. We have therefore modified the introduction and the abstract of the manuscript, and rephrased these parts to make our point clear. There is a key conceptual difference between the phase diagram of antiferromagnets and of ferromagnets as a function of applied magnetic field and temperature, which is important when interpreting magnetotransport measurements. That is what we had in mind when writing our text, and what we now explain much more explicitly in the introduction of the resubmitted manuscript.

The key difference is the following. In antiferromagnets, the magnetoconductance (or its derivative relative to the magnetic field) exhibits sharp features when passing across phase boundaries in the H-T phase diagram. These phase boundaries are continuous lines originating typically from spin-flip or spin-flop transitions (details depend on the nature and size of exchange energy and of anisotropy energy), across which the symmetry of the magnetic state changes. That is: passing across a phase boundary the system undergoes a true phase transition. In ferromagnets the situation is different. At zero applied magnetic field ($H=0$) there is a true phase transition at the critical temperature T_c . For $T>T_c$ the system has zero magnetization ($M=0$) and is in a high symmetry state; for $T<T_c$ the magnetization is finite and the symmetry is lower. At any finite magnetic field, however, the magnetization M is strictly speaking finite at all temperatures, and the symmetry is always low. In the

presence of an applied magnetic field, therefore, there is strictly speaking no phase transition separating the ferromagnetic and the paramagnetic states. The critical point at $H=0$ and $T=T_c$ is then isolated and not part of a critical line in the H - T plane. For this reason, it is less clear what the magnetoconductance in a ferromagnet really measures.

In practice, as it is done for different systems, one can follow a feature in the magnetoconductance data and heuristically define a line that separates the paramagnetic and the ferromagnetic states. However, since in the presence of an applied field the paramagnetic and the ferromagnetic state are not really distinct, this separation does not correspond to a true phase transition. In this context, we claim that our results are important because they bypass this situation, by relating the magnetoconductance data to a well-defined physical quantity, namely the magnetization.

Having explained this key conceptual issue, which we now elaborate on explicitly in the introduction of our manuscript, we need to say that we do not agree with the statement of the reviewer that *“In a magnetic tunnel barrier, be it ferromagnetic or antiferromagnetic, the tunneling conductance is expected to be dependent on magnetization...”*. We have worked ourselves on CrI_3 , CrCl_3 and MnPS_3 barrier, and we can say that this statement is not factually correct in general. It is true that in all cases the magnetoconductance depends on the magnetic state of the system, but this does not mean that it depends on the magnetization. For instance, in CrI_3 flipping the magnetization of the layers deep inside in a tunnel barrier causes a much larger change in conductance than flipping the magnetization in the surface layer, despite the change in overall magnetization being the same in the two cases (see for instance Ref. 7 of our manuscript). This example by itself shows that the magnetoconductance is not, in general, a function of the magnetization. In MnPS_3 , past the spin-flop transition at ~ 5 T, the magnetoconductance decreases upon increasing the applied magnetic field (and hence also the field-induced magnetization), exhibiting a trend opposite to the one we revealed in CrBr_3 .

In the resubmitted version of our manuscript, we now emphasize these points as part of our discussion of the interpretation of the experiments. Comparison of tunneling magnetoconductance as a function of H and T in the different materials investigated indicates that a positive magnetoconductance that is function of the magnetization is a distinctive signature of tunneling through a ferromagnetic barrier, and known antiferromagnetic barriers do not show this behavior.

Comment 3: *The main conclusion of the paper, as highlighted by the authors on page 5, last paragraph, “magnetoconductance measurements can be used to investigate the magnetic properties of the material.” and “magnetoconductance measurements will allow detailed investigations of the critical behavior of the magnetic susceptibility in the paramagnetic state for T very close to T_C and provide a new, experimentally simple way to determine critical exponents”. These need to be highlighted more in the abstract of the paper.*

Reply 3: We understand the reason why the reviewer makes this comment and we agree. We have largely modified the abstract as compared to that of the originally submitted version.

Comment 4: *Page 1- paragraph 3- “These conclusions have been drawn from experiments on different antiferromagnetic insulators (CrI_3 [7–9], CrCl_3 [10–13] and MnPS_3 [15]), and it is not at all clear that the technique can be equally effective to probe ferromagnetic insulators since in that case, no magnetic phase boundaries are present below the Curie temperature. Indeed, tunneling conductance measurements on ferromagnetic CrBr_3 barriers have shown only an extremely small and featureless low-temperature tunneling magnetoconductance^{13,14,17”} Whereas I agree, that in previous literature, tunneling conductance measurements on ferromagnetic*

CrBr₃ barriers have shown only an extremely small and featureless low-temperature tunneling magnetoconductance, I argue that the experimental data for CrBr₃ presented in (Nano Lett. 2019, 19, 5739, reference 14 in the manuscript) has similar behavior compared to the present manuscript, and show that tunneling measurements allow for an easy determination of the field–temperature phase diagram.

Reply 4: We certainly agree that we are not the first to perform systematic magnetotransport measurements on CrBr₃. As appreciated by this referee, as well as and by the others, the novelty of our work is not in the measurements themselves, but in the analysis and in the physical understanding that originates from it.

To avoid misunderstandings, in the introduction of the resubmitted version of our manuscript we also have added a paragraph in which we overview explicitly earlier experimental results obtained through magnetotransport measurements.

Comment 5: *Fig. 1c bottom right inset: It is more appropriate to plot this data as a function of $1/V$, as d is just a constant for both the curves. Plotting as a function of $1/V$ also makes it clear above which voltage the tunnel current turns on.*

Reply 5: We have modified the figure as suggested by the referee.

Comment 6: *Fig. 1c main panel: it will be good to highlight which point in the I/V curve was chosen to plot 1e. Naively, from eye-estimate 1 V and 1.4 V seems to lie just above the turning point. How does the G - T curve change if a higher current value is chosen?*

Reply 6: We have followed the suggestion of the referee by adding two stars on the curve, which also illustrate that our magnetotransport measurements were done in the Fowler-Nordheim tunneling regime. We have measured magnetoconductance with different voltage bias at fixed temperature, to verify that the voltage bias does not affect the general behavior of magnetotransport. We did not measure the G - T curve with different current, but the current levels that we have used are in the range of those used in Reference 15, where it was shown that bias current do not influence the behavior of temperature dependent conductance.

Comment 7: *Which precautions were taken to avoid Joule heating while performing low-temperature measurements?*

Reply 7: We intentionally use small current to avoid Joule heating (the dissipated power is below 5 nW in all our measurements), in the range shown not to cause heating problems in Ref. 15.

Comment 8: *Fig. 1d): δG - T shows a weak-localization behavior. Can this be attributed to the graphite leads?*

Reply 8: No, the resistance of the graphene leads is negligible with respect to that of the tunnel barrier and the effect of weak localization would not be appreciable on the scale of the measurements plotted in the figure. The magnetoresistance plotted in the figure originates from the tunnel barrier. It is small because at low temperature the spins in CrBr₃ are almost perfectly aligned at $H = 0$, and the application of a magnetic field does not influence much the magnetic state.

Comment 9: *Fig. 2 a and b): Which temperature and magnetic field intervals were used to obtain these plots?*

Reply 9: The temperature interval is 2 K/step and the magnetic field interval is 0.005 T/step, we have added this information in the caption of Fig.2.

Comment 10: *Fig 2: At which values of voltage or current were this data obtained?*

Reply 10: The bias voltage used is 1 V and 1.4 V for N=7 and N=8 devices, respectively, for all temperature and magnetic field dependence of tunneling conductance discussed in the manuscript. We have modified the caption of Fig.1 to clearly clarify this.

Comment 11: *Page 3, paragraph 2: “We can therefore conclude directly from the data that the magnetoconductance δG depends on H and T only through the magnetization $M(H; T)$ even well outside the linear regime. That is: for $T > T_C$, $\delta G(H; T) = \delta G(M(H; T))$ ”. Can the authors discuss which other factors can potentially influence magnetoconductance in these systems, that are eliminated by this data? For example, CrBr₃ is known to influence Hall measurements in graphene through proximity coupling (Nature Electronics volume 2, pages457–463 (2019)).*

Reply 11: As we discussed above, the magnetoconductance of magnetic materials in general depends on their magnetic state, and this does not mean that it has to depend on (or only on) the magnetization. For instance, it could depend on the magnitude of spin fluctuations (e.g., on the magnetic susceptibility) and not only on the average spin (that is on the magnetization). We now mention this explicitly in the manuscript. One mechanism that could lead to such a dependence in tunneling transport is dephasing/decoherence: it is well known from past work in mesoscopic physics that enhancing decoherence tends to strongly suppress tunneling. If that mechanism was dominant, scattering on fluctuating magnetic moments could have a dominating effect on tunneling conductance. Our results show that this is not the case.

As for the influence of CrBr₃ on graphene that may certainly be there, but since the resistance of the graphene leads is entirely negligible, this would not affect what we see in our devices.

Comment 12: *Page 3, paragraph 4: “exfoliated atomically thin crystals of the size used in our devices have been found to behave as single domains”. The movement of domain walls was previously reported in monolayer CrBr₃ (Nature Electronics volume 2, pages457–463 (2019)).*

Reply 12: We agree that what we wrote about thin crystals being single domain is unprecise and we corrected our statements. The point of the discussion in which these statements were made was to explain that the M - H curve measured in bulk crystals is not representative of the curve measured in atomically thin crystals. Indeed, in bulk crystals virtually no hysteresis is observed in the magnetization versus field and the remnant magnetization at zero applied field vanishes. In atomically thin crystals the situation is different. A clear hysteresis is observed in the M - H curve and there is a finite remnant magnetization that depends on temperature. The difference between bulk and atomically thin crystals is rooted in the physics of domains, but it is incorrect to state that atomically thin layers are single domain in all cases.

We have rephrased the discussion referring to hysteresis and remnant magnetic field, in a way that does not rely on interpretation and that is factually correct.

Comment 13: *Is the data shown in Fig. 3 obtained in the 7-layer device? Is Fig. 3a just a repetition of Fig. 2b?*

Reply 13: Yes it is. We want to include the data also in Figure 3, because it makes much simpler to understand what we do in practice to plot the magnetoconductance as a function of magnetization.

Comment 14: *“Crystals of CrBr₃ were grown by the Chemical Vapour Transport method as reported earlier.” Reference is missing here.*

Reply 14: We thank the reviewer for pointing it out and we have added the reference.

Comment 15: *It is not absolutely necessary, but it will be nice to see images of grown crystals in the supporting information.*

Reply 15: We followed the suggestion of reviewer and added an image of a grown crystal in the supplementary information.

Comment 16: *According to the images shown in supplementary Fig. 2: The heterostructures have quite a few bubbles trapped. Can the authors show a closer image of the tunnel junction area, to demonstrate no bubbles are present in that area? Or, even if the bubbles are present, can the authors comment on how the bubbles are going to affect the magnetoconductance behavior?*

Reply 16: The bubbles are not in the tunneling area and do not connect two graphene electrodes. The virtually perfect reproducibility of the results that we obtained in the 7-layer and 8-layer devices show that the behavior that we observed is intrinsic and not affected by spurious effect.

Comment 17: *Along with the above I also have a few minor comments:*

- 1. In formal scientific writing “em dash” is not used widely. I request the authors to minimize the use of em-dash. In addition, the authors are requested to note that, no space is required before or after emdash.*
- 2. Page 1 paragraph 4 line 13: “scheme” will be “schematic”*
- 3. Page 5 paragraph 1: “This is illustrated by the color plot in Fig. 4d” and “compare Fig. 4d with Fig. 2a”- Here “4d” will be “4c”*
- 4. Fig. 1: “insert” will be “inset”*

Reply 17: We have corrected this type of errors in the main text.

Reply to Reviewer #4:

Comment 1: *The authors present experimental results on the tunneling magnetoconductance measurements across graphene/CrBr₃/graphene devices. They analyze the magnetoconductance using the conductance at zero magnetization state ($H=0$, $T \geq T_c$) as references. They find that the behavior of the normalized magnetoconductance solely depends on the magnetization of CrBr₃. This behavior can be explained by the Fowler-Nordheim model by taking into account the magnetization dependent tunnel barrier heights for the two spin channels. These results are interesting and useful for the 2D magnet community. I would recommend publication in Nature Communications. Below I have a few minor comments.*

Reply 1: We are pleased to see that the reviewer finds the results reported in our manuscript interesting and useful for the community and thank the reviewer for the unconditional recommendation of publication. We have taken into account the minor comments indicated by the referee in the resubmitted version of our manuscript.

Comment 2: *When analyzing the data below T_c , the authors use the XXZ model to calculate the expected M values instead of using the experiment results due to the domain formation at small fields (1 mT). Would it be possible to use the measured M values at slightly higher field, large enough to polarize the domain but small enough to induce any appreciable change in M , for example, 0.1 T? Also, the authors should provide a short description of the model calculation.*

Reply 2: In analyzing the data we have just extracted the results from the paper from the Manchester group (Reference 24). We now make this clear in the manuscript. As a note, we have seen that the M -vs- T curve in that paper matches basically perfectly, at $H=0$, to the common mean field Weiss model of ferromagnetism (differences between the Weiss model and the XXZ model may appear at finite field, but we have not analyzed the XXZ model as a function of field and we are not able to tell).

As for using the value of M at small field, there is no saturation in bulk magnetization up to at least 0.3 T (supplementary Fig. 2), which is already a pretty large field and affects the T dependence. So, in practice we do not have suitable data to implement the suggestion of the referee, which is why for our analysis we resorted to published magnetization measurements performed on atomically thin crystals.

Comment 3: *Fig. 3c is calculated based on fitting parameters obtained from the experimental data. But the maximum ΔG is only 80% while the maximum in experiment is 130% (Fig. 2a). What causes such a big difference?*

Reply 3: We believe that the referee actually refers to Fig. 4c and not 3c. As we mentioned in the text, the results plotted in Fig. 4c rely on the use of the simplest Weiss model of magnetism. This model does not exactly reproduce the magnetization curves of CrBr₃, which is the reason for the quantitative deviation. Nevertheless, despite relying on the simplest possible model, the approach that we have followed reproduces all qualitative aspects of the data. This is what we want to illustrate with the plot. A more sophisticated model of magnetism may do a better job. The idea that we mention in the manuscript is that by comparing the level of agreement we can then discriminate between different microscopic Hamiltonians to determine the one that optimally describes CrBr₃.

Comment 4: *The direction of the magnetic field in the magnetoconductance measurements should be specified somewhere in the paper.*

Reply 4: We thank the reviewer for noticing that we had forgotten to mention the direction of the magnetic field. We have added this information in the caption of Fig.1 and in the main text.

REVIEWERS' COMMENTS

Reviewer #1 (Remarks to the Author):

The authors have addressed my questions and concerns. I recommend the publication.

Reviewer #3 (Remarks to the Author):

I thank the authors for their detailed comments and modification of the manuscripts. I think the modifications have significantly improved the manuscripts and the authors have satisfactorily answered all my questions.

I have a few minor follow-up comments.

1. It will be good if the authors can add the precautions they have taken to avoid Joule heating in the manuscript.
2. I request the authors to explicitly mention the bias voltage used in Fig. 2.

Reviewer #4 (Remarks to the Author):

My questions have been addressed satisfactorily. I would recommend it for publication.

Reply to referees:

We thank all the referees for their valuable comments during the reviewing process and their recommendation for the publication. Only referee 3 has two minor comments, we have followed the suggestion of the referee and modified the caption of Fig. 1 and Fig. 2 in main text.

Reviewer #1 (Remarks to the Author):

The authors have addressed my questions and concerns. I recommend the publication.

Reviewer #3 (Remarks to the Author):

I thank the authors for their detailed comments and modification of the manuscripts. I think the modifications have significantly improved the manuscripts and the authors have satisfactorily answered all my questions.

I have a few minor follow-up comments.

- 1. It will be good if the authors can add the precautions they have taken to avoid Joule heating in the manuscript.*
- 2. I request the authors to explicitly mention the bias voltage used in Fig. 2.*

Reviewer #4 (Remarks to the Author):

My questions have been addressed satisfactorily. I would recommend it for publication.